# Genome-Informed Real-Time PCR Assay for Detection of ‘*Candidatus* Phytoplasma Prunorum,’ Which Is Associated with European Stone Fruit Yellows

**DOI:** 10.3390/microorganisms13040929

**Published:** 2025-04-17

**Authors:** Jarred Yasuhara-Bell, Yazmín Rivera

**Affiliations:** Plant Pathogen Confirmatory Diagnostics Laboratory (PPCDL), Science and Technology (S&T), Plant Protection and Quarantine (PPQ), Animal and Plant Health Inspection Service (APHIS), United States Department of Agriculture (USDA), Laurel, MD 20708, USA; yazmin.rivera@usda.gov

**Keywords:** ‘*Candidatus* Phytoplasma mali’, phytoplasma, apple proliferation, detection, PCR

## Abstract

‘*Candidatus* Phytoplasma prunorum’ has been associated with severe disease in *Prunus* spp., which are commodities of economic importance in the USA. The introduction and establishment of ‘*Ca.* P. prunorum’ in the USA could result in huge economic losses, thus creating a need for validated diagnostic tools, which are the cornerstone of successful surveillance, quarantine, and eradication measures. Whole-genome comparisons led to the identification of a diagnostic marker gene specific to ‘*Ca.* P. prunorum’ (PE639). The PE639 assay was duplexed with an 18S rDNA plant internal control and compared to modified 23S (phytoplasmas) and *imp* (‘*Ca.* P. mali’) assays. The PE639 assay produced congruent results to 23S and *imp* assays for all metrics, demonstrating high linearity, repeatability, intermediate precision, and reproducibility. The limit of detection was comparable for all assays tested, and all demonstrated 100% analytical specificity, selectivity, and diagnostic specificity for their respective target species. Assays metrics were consistent across two platforms, the ABI QuantStudio™ 5 and Bio-Rad CFX96™ OPUS. A synthetic gBlocks™ control was designed and validated to work with all assays, as well as conventional PCR assays targeting 16S rDNA and *tuf* genes. These validated assays and synthetic control represent beneficial tools that support efforts to protect USA agriculture and facilitate safe trade.

## 1. Introduction

Phytoplasmas of the apple proliferation group (16SrX) can be associated with severe disease in fruit trees of the Rosaceae family. The 16SrX group contains ‘*Candidatus* Phytoplasma mali’ (16SrX-A), which is associated with apple proliferation (AP), ‘*Ca*. P. prunorum’ (16SrX-F), which is associated with European stone fruit yellows (ESFY), and ‘*Ca*. P. pyri’ (16SrX-C), which is associated with pear decline (PD) [1,2]. The primary hosts for ‘*Ca*. P. mali’ and ‘*Ca*. P. pyri’ are *Malus* spp. and *Pyrus* spp., respectively. ‘*Candidatus* P. prunorum’ is also associated with several economically relevant diseases of *Prunus* spp., which are collectively referred to as ESFY and include diseases of apricots, Japanese plums, and peaches [3].

Apples and stone fruit are important commodities in the United States of America (USA). Between 2019 and 2022, the average production of apples in the USA was ~4.7M metric tons (tonnes), with an estimated value of ~USD 3.8B, while production of stone fruit (apricots, peaches, and plums) was ~1M tonnes, with an estimated value of ~USD 1.6B [4,5]. Both ‘*Ca*. P. prunorum’ and ‘*Ca*. P. mali’ have not been detected in the USA [6,7,8,9], while ‘*Ca*. P. pyri’ is reported to occur throughout North America [10,11]; ‘*Ca*. P. mali’ has been detected in North America in Nova Scotia, Canada [6,9]. Both ‘*Ca*. P. prunorum’ and ‘*Ca*. P. mali’ are considered quarantine pathogens by the USA and Canada, as introduction of these plant pests into the USA could have significant economic consequences.

Phytoplasmas can be spread through propagation practices; therefore, long-distance spread can occur by movement of infected propagative materials, including trade of infected rootstock, scion wood, or budwood. In addition, phytoplasmas have been found infecting alternative host species [2,3,12,13,14,15,16,17,18]. Cryptic infections in asymptomatic plants and/or alternative host species can facilitate unintended introduction of phytoplasmas into new areas. Controlling the disease and spread of these quarantine organisms requires accurate and rapid detection methods. Large-scale surveillance is necessary to identify possible introductions of ‘*Ca*. P. prunorum’ and ‘*Ca*. P. mali’ and to deploy management strategies quickly to eradicate the pathogens before they become established.

‘*Candidatus* Phytoplasma prunorum’ and ‘*Ca*. P. mali’ are USDA Animal and Plant Health Inspection Service (APHIS) Plant Protection and Quarantine (PPQ) Cooperative Agricultural Pest Survey (CAPS) Program priority pests. The CAPS Program Approved Method for Pest Surveillance for ‘*Ca*. P. prunorum’ and ‘*Ca*. P. mali’ involves collecting symptomatic plant tissue by visual survey (3031-General Visual Observation) and then sending samples to screening labs for molecular testing. The approved molecular screening assay was validated by the USDA APHIS PPQ Science and Technology Plant Pathogen Confirmatory Diagnostic Laboratory (S&T PPCDL) and is a real-time PCR assay designed to detect all phytoplasmas; it targets the 23S rDNA genes [19] and was modified to include a host plant 18S rDNA internal control [13].

As a result of host overlap with native phytoplasmas, species-specific detection is necessary to specifically identify these regulated pathogens. Real-time PCR is often preferred over conventional PCR (cPCR) for use as a screening tool due to its increased throughput, reduced and simplified steps, reduced time, and relatively straightforward data analysis. Real-time PCR assays for specific detection of ‘*Ca*. P. prunorum’ [20,21,22,23,24,25,26,27] and ‘*Ca*. P. mali’ [22,23,28,29,30,31,32,33,34] have been described previously in the literature. TaqMan-based real-time PCR is preferential to SYBR-based assays due to increased specificity and the ability to multiplex assays, particularly with a host internal control that can provide well-to-well assessment of sample extraction quality and reagent functionality. Many of the published assays are SYBR-based, and the majority of their targets are taxonomic markers (16S rRNA, internal transcribed spacer (ITS), ribosomal proteins (*rpl22*), and immunodominant membrane protein (*imp*)) that exist in all phytoplasmas and can be similar among the 16SrX group; an *imp* assay was validated and shown to be specific to ‘*Ca*. P. mali’ [34,35].

Recently, PPCDL sequenced and assembled draft genomes for two ‘*Ca*. P. prunorum’ strains (ESFY1 and LNS1), which are now available in the GenBank public database [36]. Given the availability of this new resource, the aim of this project was to develop and validate a screening tool by identifying a genome-informed diagnostic marker that can specifically detect ‘*Ca*. P. prunorum,’ to aid in surveillance and detection efforts to safeguard USA agriculture and facilitate safe trade. The *imp* assay for specific detection of ‘*Ca*. P. mali’ [34,35] and 23S rRNA assay [19] for general detection of phytoplasmas were also included in the validation testing performed in this study.

## 2. Materials and Methods

### 2.1. Strains, Plant Hosts, and DNA Extraction

DNA extracts for 32 phytoplasma strains, including 11 ‘*Ca*. P. prunorum’ and eight ‘*Ca*. P. mali’ strains, were acquired from various sources (Table 1); most were purchased from the EPPO-Q-bank Phytoplasma Collection at the University of Bologna, Italy. Additionally, 33 ‘*Ca*. Phytoplasma’-positive diagnostic samples from a variety of hosts were included in this study (Table 1). DNA was extracted using 200 mg of tissue and the DNeasy Plant Mini Kit (Qiagen, Germantown, MD, USA), with a modified protocol to include CTAB (cetyltrimethylammonium bromide) extraction buffer (Promega, Madison, WI, USA) and 2-mercaptoethanol instead of the kit-provided AP1 extraction buffer, or the DNeasy Plant Pro Kit (Qiagen). DNA from 60 additional organisms, comprising a few biocontrols and many pathogens affecting apples, pears, and stone fruit, were acquired from various providers and included in this study for exclusivity testing (Table 1).

Healthy leaf tissue from Asian pear (*Pyrus pyrifoliae*), apple (*Malus domestica*), European pear (*Pyrus communis*), apricot (*Prunus armeniaca*), cherry (*Prunus avium*), and Japanese plum (*Prunus salicina*) trees were provided by the USDA APHIS PPQ Plant Germplasm Quarantine Program (PGQP). Fresh strawberries (*Fragaria* × *ananassa*) (Driscoll’s, Watsonville, CA, USA) and freeze-dried strawberry fruit (Brothers All Natural, Rochester, NY, USA) were purchased from a local grocer. DNA was extracted from healthy plant tissue using the DNeasy Plant Mini Kit. DNA from peaches (*Prunus persica*) and plums (*Prunus* sp.) were provided by Y.K. Jo at Texas A&M University, and DNA for Asian pears (*Pyrus pyrifoliae*; var. Kosui, Kumoi, Hayatama and Tama) and Chinese pears (*Pyrus ussuriensis*; var. China and Mien Suan Li) were provided by N. Bassil and R. King at USDA ARS (Agricultural Research Service) National Clonal Germplasm Repository (NCGR) in Corvallis, Oregon.

### 2.2. Target Identification and Primer Design

Whole-genome sequences for ‘*Ca*. P. prunorum’ strain ESFY1 (JAZHCZ000000000.2) [36], ‘*Ca*. P. prunorum’ strain LNS1 (JAZHCY000000000.2) [36], aster yellows witches-broom phytoplasma strain AYWB (GenBank accession number CP000061.1) [38], ‘*Ca*. P. australiense’ (GenBank accession number AM422018.1) [39], ‘*Ca*. P. mali’ strain AT (CU469464.1) [40], ‘*Ca*. P. solani’ strain 284/09 (FO393427.1) [41], onion yellows phytoplasma strain OY-M (GenBank accession number AP006628.2) [42], and strawberry lethal yellows phytoplasma (CPA) strain NZSb11 (GenBank accession number CP002548.1) [43] were acquired from the National Center for Biotechnology Information (NCBI) GenBank. Genomes were annotated and compared to identify orthologous genes using the MicroScope Microbial Genome Annotation and Analysis Platform version 3.16.1 [44,45,46,47,48,49] under stringent parameters (80% amino-acid identity and 80% amino-acid alignment coverage).

Primers and probes were designed to the PE639 sequence using the PrimerQuest™ tool from Integrated DNA Technologies (IDT, Coralville, IA, USA) (https://www.idtdna.com/SciTools) [50] (accessed on 29 September 2023) and evaluated using NCBI Primer BLAST (https://blast.ncbi.nlm.nih.gov/Blast.cgi) [51] (accessed on 29 September 2023) to assess matches with non-targets, with default parameters. Primer and probe binding sites were mapped to the target sequence in Geneious Prime^®^ 2023.0.4 [52], and the target amplicon was extracted and rescreened using NCBI BLASTn [51] (megablast and blastn with default parameters) to assess matches to non-targets; whole-genome shotgun contigs (wgs), core nucleotide (core_nt), Refseq Genome (refseq_genomes), and Refseq Reference Genome (refseq_reference_genomes) databases were queried. The PE639 primer and probe sequences are shown in Table 2.

A modified real-time PCR specific to ‘*Ca.* P. mali’ (a closely related 16SrX group phytoplasma) targeting the *imp* gene [34,35] was also included in this study, along with a modified real-time PCR assay targeting 23S rDNA for genus-level phytoplasma detection [19] (Table 2). Assay modifications included the use of different primer/probe concentrations, different fluorophore-quencher pairing, different master mix and reaction conditions, and inclusion of an internal control; each primer set was duplexed with an internal control (IC) primer set targeting 18S rDNA of plants [13] (Table 2). All assays are hereinafter referred to by their gene target names (PE639, *imp*, 23S, 18S, and 16S).

### 2.3. PCR Reaction Conditions

Duplex real-time PCR reactions contained 1X PerfeCTa^®^ qPCR ToughMix^®^ Low ROX^™^ (Quantabio, Beverly, MA, USA), a target and 18S IC primer/probe set (final concentrations presented in Table 2), and 2 µL of DNA template in a final reaction volume of 25 µL. Reactions were run in the QuantStudio™ 5 Real-Time PCR System (Applied Biosystems, Waltham, MA, USA) according to the following cycling conditions: initial denaturation at 95°C for 2 min, followed by 40 cycles of 95°C for 5 s and 58°C for 40 s. A ramp rate of 1.60°C/s was used, and manual thresholds were set at 0.23 and 0.09 for the targets and 18S IC, respectively. Assays for ‘*Ca*. P. prunorum’ and ‘*Ca.* P. mali’ were also run in a CFX Opus Real-Time PCR System (Bio-Rad, Hercules, CA, USA) using the reaction conditions stated above. The 18S (ABY-QSY) fluorophore/quencher pair was changed to VIC-QSY (Applied Biosystems) and Cy3-IAbRQSp (Iowa Black^®^ RQ; IDT) when duplexed with PE639 (‘*Ca*. P. prunorum’) and *imp* (‘*Ca.* P. mali’), respectively, to ensure compatibility with the CFX96 OPUS. The ‘*Ca.* P. mali’ assay was also run in a CFX96 Touch Real-Time PCR Detection System (Bio-Rad) under the same parameters. Manual thresholds for the CFX96 instruments were set at 400 for FAM targets, and 350 and 50 for 18S ICs with VIC and Cy3, respectively. Modified 23S and *imp* assays showed equivalent or better performance to the original assays.

Semi-nested cPCR reactions for detection and identification of phytoplasmas contained 1X AccuStart II PCR SuperMix (Quantabio), 400 nM of each primer (P1/16S-SR for the 1st round; P1A 16S-SR for the 2nd round) (Table 2) [53,54], and 2 µL of DNA template in a final reaction volume of 50 µL. Reactions were run in the ProFlex PCR System (Applied Biosystems), or equivalent thermocycler, according to the following cycling conditions: initial denaturation at 95°C for 5 min, followed by 38 cycles of denaturing at 95°C for 15 s, annealing at 55°C for 30 s, and elongation at 72 °C for 90 s, and a final extension at 72°C for 4 min; maximum ramping was used. Amplicons from the 1st round were diluted 1:30 for use as a template for the 2nd round of the semi-nested cPCR.

Nested cPCR reactions for amplification of the *tuf* DNA barcode were carried out as published previously [55]. Alternatively, AccuStart II PCR SuperMix was used to replace the individual reagent components of the master mix. Reactions were 25 µL in volume and contained 1 µL of sample. Two sets of primer cocktails were used for each round of nested cPCR: Tuf340/Tuf890 for the 1st round and Tuf400/Tuf835 for the 2nd round (Appendix A). Reactions were run in the ProFlex PCR System, or equivalent thermocycler, according to the following cycling conditions: initial denaturation at 94°C for 3 min, followed by 35 cycles of denaturing at 94°C for 15 s, annealing at 54°C for 30 s, and elongation at 72°C for 1 min, and a final extension at 72°C for 7 min. Amplicons from the 1st round were diluted 1:30 for use as a template for the 2nd round of the nested cPCR.

### 2.4. Gel Electrophoresis and Sequencing

16S PCR products were analyzed using the 4200 TapeStation System, D5000 ScreenTape, D5000 DNA Ladder, D5000 Sample Buffer, and 4200 TapeStation Controller Software (version 5.1) (Agilent Technologies, Inc., Santa Clara, CA, USA), while *tuf* amplicons were analyzed using the D1000 ScreenTape, DNA Ladder, and Sample Buffer, according to manufacturer’s instructions. Amplicons were cleaned using ExoSAP-IT™ Express PCR Product Cleanup Reagent (Applied Biosystems), according to manufacturer’s instructions. Cleaned PCR products were sequenced using a SeqStudio™ Genetic Analyzer (Applied Biosystems), BigDye™ Terminator v3.1 Cycle Sequencing Kit (Applied Biosystems), and BigDye XTerminator™ Purification Kit (Applied Biosystems), according to the manufacturer’s recommendations.

### 2.5. Synthetic Positive Control

A synthetic gBlocks™ (IDT) positive control was designed for use with all described PCR assays. As shown in Appendix A, the synthetic positive control has a base sequence comprising 1793 bp of 16S from strawberry lethal yellows phytoplasma (CPA) str. NZSb11 (GenBank accession number CP002548) and contains all primer binding sites for the 16S semi-nested cPCR. The base sequence was modified to include phytoplasma 23S (149 bp) and plant 18S (68 bp) amplicons for real-time PCR by replacing the existing sequence; inclusion of 18S sequences as an internal control to simulate a plant host obviated the need to spike the control into plant host material/DNA. PE639 (139 bp) and *imp* (88 bp) amplicons for ‘*Ca*. P. prunorum’ and ‘*Ca.* P. mali,’ respectively, were also inserted by replacing existing base sequence. Additionally, *tuf* gene primer binding sites were inserted by replacing existing base sequence, allowing use of the synthetic control for *tuf* gene sequencing when trying to increase discrimination between closely related taxa. Two unique identifiers were incorporated into the synthetic gBlocks: (1) a 16S target region was intentionally replaced by the 23S target so that this control would test negative when using the assay described by Christensen et al. [13]; (2) base sequence was modified so that the translated amino acid sequence will read “APHIS*SCIENCE” when viewing the first frame. The complete 1793 bp sequence of synthetic gBlocks™ positive control is shown in Appendix A. The synthetic control was used at a concentration of 4 fg/µL; working stocks were made in TE buffer with 0.5 mg/mL polyadenylic acid (poly(A)) (Roche Diagnostics Deutschland, Mannheim, Germany).

### 2.6. Assay Validation Testing

Validation testing evaluated the analytical specificity and sensitivity, diagnostic specificity, repeatability, and reproducibility of the optimized diagnostic assays. DNA from ‘*Ca.* P. prunorum’ strains was used to determine analytical specificity, more specifically, inclusivity (Table 1). DNA extracted from healthy plant species, non-16SrX-F group phytoplasmas, and other pathogens of apples, pears, and stone fruit were used to assess exclusivity (selectivity) (Table 1). Diagnostic specificity was assessed using archived DNA extracted from diagnostic samples collected in the USA previously and determined positive for non-16SrX-F group phytoplasmas (Table 1); samples did not contain ‘*Ca.* P. prunorum’ DNA and were expected to produce negative results (FAM C_t_ = 0/Undetermined).

DNA extracted from ‘*Ca*. P. prunorum’ strains ESFY1, LNp, and LNS1 were diluted in healthy peach (*Prunus persica*) DNA initially to produce 23S-FAM C_t_ ~25; this was designated Dilution 0 (10^0^). Ten-fold serial dilutions were then made to determine assay analytical sensitivity; dilutions were made in healthy peach (*Prunus persica*) DNA to achieve a constant 18S-ABY C_t_ ~20. The determined limit of detection (LoD) was further tested in different host backgrounds. Mock samples were produced by spiking DNA extracted from healthy apricots (*Prunus armeniaca*), cherries (*Prunus avium*), and Japanese plums (*Prunus salicina*) with DNA extracted from ‘*Ca.* P. prunorum’ strains ESFY1, LNp, and LNS1 at the LoD concentration.

Repeatability was assessed across the dilution series using data from a single operator (N = 3). Intermediate precision (N = 9) and reproducibility (N = 12) were evaluated using data from two additional operators from a different laboratory team at USDA APHIS PPQ S&T PPCDL. Pre-tested primers, probes, and positive controls were provided to participants, along with a copy of the testing protocol. Each participant tested serial dilutions of DNA extracted from ‘*Ca.* P. prunorum’ strain ESFY1 in healthy peach (*Prunus persica*) DNA; samples were tested in triplicate. A set of serial dilutions of ‘*Ca.* P. prunorum’ strain ESFY1 in healthy peach (*Prunus persica*) DNA and ‘*Ca.* P. mali’ strain C71 in healthy apple (*Malus domestica*) tested in the QuantStudio™ 5 instrument previously were used to validate the assays in the CFX96 OPUS (PE639 and *imp* assays) and CFX Touch (*imp* assay only).

## 3. Results

### 3.1. Target Identification

Based on the analysis described above, 115 genes were identified as being present in the two ‘*Ca.* P. prunorum’ strains and absent in the other ‘*Ca*. Phytoplasma’ species. Identified genes were checked for specificity in silico using NCBI BLASTn [51] (megablast and blastn with default parameters) to assess matches to non-targets; whole-genome shotgun contigs (wgs), core nucleotide (core_nt), Refseq Genome (refseq_genomes), and Refseq Reference Genome (refseq_reference_genomes) databases were queried. A 639 nt putative effector protein (MICFAM_ID 18335 [44,45,46,47,48,49], designated as PE639) was identified from these 115 genes as useful diagnostic marker, matching only to ‘*Ca.* P. prunorum’ sequences (locus tags V9D73_RS02055 and V9D86_RS01530 for ESFY1 and LNS1, respectively). Genome sequences for ‘*Ca.* P. prunorum’ strains ESFY1 and LNS1 were input into Geneious Prime^®^ 2023.0.4 [52], and PE639 sequences were extracted, aligned, and showed 100% identity among strains.

### 3.2. Assay Sensitivity

The LoD was determined using serial dilutions of ‘*Ca.* P. prunorum’ strains ESFY1, LNp, and LNS1 in healthy peach (*Prunus persica*) DNA, including aggregate data from three operators and two instruments for ESFY1; PE639 assay data were compared to 23S assay data. On the QuantStudio™ 5, the LoD (lowest dilution with 100% positivity) was determined to be Dilution 3 (10^−3^) for ESFY1 and LNp, which corresponded to C_t_ values between 35 and 36 (Appendix A); the LoDs for PE639 and 23S were congruent. Based on previous results from testing serial dilutions of plasmids containing the 23S rRNA amplicon with the 23S assay, the LoD was calculated to be ~2–20 genome equivalent copies (data not presented). For LNS1, C_t_ values for corresponding dilutions were approximately 10-fold lower for PE639 when compared to 23S; the LoD for PE639 was Dilution 4 (10^−4^), while the LoD for 23S was Dilution 3 (10^−3^), both corresponding to C_t_ values between 34 and 36 (Appendix A). On the CFX96 OPUS, data for dilutions of ‘*Ca.* P. prunorum’ strain ESFY1 DNA revealed the same LoD as the QuantStudio™ 5 (Appendix A). Additional tests were performed to assess assay performance at the LoD using DNA from ‘*Ca.* P. prunorum’ strains ESFY1, LNp, and LNS1 spiked into DNA extracted from different hosts (Appendix A); the 23S assay was evaluated using LNS1 only. Data showed that PE639 can detect ‘*Ca.* P. prunorum’ DNA with 100% positivity at the LoD in a background of multiple other hosts (apricot, Japanese plum, and cherry). The IC produced C_t_ values within the expected C_t_ range for the 18S target for all tests.

Diagnostic samples may require sequence-based identification in addition to detection by real-time PCR. Therefore, cPCR for 16S and sequencing were performed on ESFY1 and LNS1 DNA at the LoD, as well as a 10-fold higher dilution (LoD^−1^), to determine if samples near the LoD (low-titer samples) can be confirmed by downstream methods. Control reactions included the synthetic gBlocks™ positive control and a no-template control (NTC). The results showed that appropriate bands (~1500 bp) were produced for all samples tested, including the synthetic positive control (Figure 1); no bands produced for the NTC. Amplicons for ESFY1 at LoD^−1^ were subjected to Sanger sequencing and high-quality sequence data were obtained. BLAST analysis revealed matches to ‘*Ca.* P. prunorum’ sequences with >99% identity and 100% query coverage, including a 99.44% identity match to the reference sequence from ‘*Candidatus* P. prunorum’ strain ESFY-G1 (GenBank accession number AJ542544). BLAST analysis also revealed matches to matches to ‘*Ca.* P. mali’ and ‘*Ca.* P. pyri’ sequences with less than 99% but greater than 98% identity and 100% query coverage; ‘*Ca.* P. mali’ and ‘*Ca.* P. pyri’ are closely related and represent subclades of the 16SrX group. The results demonstrate that low-titer samples around the LoD of the real-time assay can be confirmed with the semi-nested 16S cPCR and sequencing.

### 3.3. Assay Specificity

Specificity and selectivity were tested using DNA isolated from known ‘*Ca.* P. prunorum’ strains, as well as other non-16SrX-F group phytoplasmas, other pathogens commonly found on the same hosts, and healthy hosts. Validation of PE639 and *imp* targets revealed 100% analytical specificity for ‘*Ca.* P. prunorum’ and ‘*Ca.* P. mali,’ respectively, as all ‘*Ca.* P. prunorum’ and ‘*Ca.* P. mali’ strains produced positive results with their respective assays (Appendix A). The 23S target also demonstrated 100% analytical specificity for ‘*Ca.* Phytoplasma,’ as all phytoplasma strains produced positive results (Appendix A). The PE639, *imp* and 23S assays only amplified DNA from ‘*Ca.* P. prunorum,’ ‘*Ca.* P. mali,’ and ‘*Ca.* Phytoplasma,’ respectively (100% diagnostic specificity); no cross-reactions were observed for PE639 and *imp* with other phytoplasmas. Additionally, these three assays did not amplify DNA from 60 non-phytoplasmas, nor with 16 plant host DNA (100% selectivity). Tests performed to assess assay performance at the LoD in different host DNA backgrounds suggest host matrices do not have a negative impact on assay performance. A small-scale specificity test was performed using the CFX96 OPUS, and data corroborated assay specificity across two instruments.

### 3.4. Precision and Reproducibility

Repeatability was assessed using serial dilutions of DNA from ‘*Ca.* P. prunorum’ strains ESFY1, LNp, and LNS1 in healthy peach (*Prunus persica*) DNA (N = 3 per data point). The PE639 assay resulted in 100% repeatability (100% positive samples at ≥LoD) and a low coefficient of variation (<2% CV) (Appendix A). Intermediate precision was assessed using serial dilutions of ‘*Ca.* P. prunorum’ strain ESFY1 in healthy peach (*Prunus persica*) DNA. Dilutions were tested by three independent operators in different QuantStudio™ 5s (N = 9 per data point). The results demonstrated high intermediate precision, with a low average variance of 1.1% across multiple operators and QuantStudio™ 5s (Appendix A); calculations included only data from dilutions with 100% positivity. Reproducibility was assessed by accounting for data produced from three operators and two instruments (QuantStudio™ 5 and CFX OPUS) (N = 12 per data point) (Appendix A). The average variance across four 10-fold serial dilutions having 100% positivity was <1.2%. The IC produced C_t_ values within the expected C_t_ range for the 18S target for all dilutions and for all tests performed.

Reproducibility data for ‘*Ca.* P. prunorum’ strain ESFY1 and ‘*Ca.* P. mali’ strain C71, as well as repeatability data for ‘*Ca.* P. prunorum’ strains LNp and LNS1 and ‘*Ca.* P. mali’ strains P4 and P6, were used to generate linearity plots (Figure 2 and Appendix A) and calculate reaction metrics (Appendix A). Data revealed high R^2^ values (>0.99) and amplifications efficiencies ≥~90% for all assays and for all strains tested (Appendix A); amplification efficiency varied depending on the strain tested. Linearity metrics were not determined for the 18S IC, as it targets host DNA, which was held constant in the diluent; all dilutions produced expected 18S IC C_t_ values. A linearity plot generated from data produced using only CFX96 OPUS data is shown in Appendix A.

### 3.5. Synthetic Control

Serial dilutions of synthetic gBlocks™ positive control, from 4 pg/µL to 400 ag/µL, were tested using all assays in both the QuantStudio™ 5 and CFX96 OPUS. The data revealed the synthetic positive control tested positive with all assay targets (PE639, *imp*, 23S, and 18S) and for all dilutions. Based on the obtained data, 4 fg/µL was selected as the working dilution. The working dilution was subjected to multiple rounds of freeze–thaw and tested by three operators in the QuantStudio™ 5 using PE639 and 18S targets. The average variance among successive rounds of testing, following intervals of freeze–thaw, was <1% for individual operators (Appendix A). Additionally, variance (“total”) was calculated to account for all data per operator (N = 15), as well as all test data (N = 45); “total” variance was also <1%, suggesting good stability and high reproducibility. The working dilution after five rounds of freeze–thaw was able to produce a bright band by cPCR (Figure 1) and produce high-quality sequences. The “APHIS*SCIENCE” unique identifier was apparent during sequence analysis. The consensus sequence was also subjected to BLAST analysis. Due to the inclusion of additional targets and unique markers within the 16S base sequence, BLAST analysis revealed highest matches to strawberry lethal yellows phytoplasma (CPA) str. NZSb11, complete genome (GenBank accession number CP002548), and ‘*Candidatus* Phytoplasma australiense’ complete genome (GenBank accession number AM422018), both with 90.50% identity and 78% query coverage. The synthetic control working dilution was also subjected to nested *tuf* cPCR and produced bands of appropriate size for each and across both rounds of PCR (Appendix A). Nested *tuf* amplicon sequences matched strawberry lethal yellows phytoplasma (CPA) str. NZSb11 and related phytoplasma 16S sequences (*e.g.*, strawberry virescence phytoplasma isolate 3101 16S ribosomal RNA gene, partial sequence; GenBank accession number AY377868), as opposed to Elongation Factor Tu (*tuf*) sequence.

## 4. Discussion

‘*Candidatus* P. prunorum’ and ‘*Ca.* P. mali’ are associated with disease of several economically important USA crops. The movement of infected plant material is the most probable means of entry, and multiple open pathways of entry exist. These pathogens are part of USA surveillance programs, and given cryptic host species, asymptomatic carriers, and overlapping hosts, there is a critical need for a validated diagnostic tool able to detect and discriminate ‘*Ca.* P. prunorum’ and ‘*Ca.* P. mali’ from domestic phytoplasmas. In this study, PPCDL addressed this need by providing validated screening tools specific to ‘*Ca.* P. prunorum’ and ‘*Ca.* P. mali.’ A literature search was performed initially to find real-time PCR assays designed for specific detection of both pathogens that could be assed at PPCDL for regulatory screening.

Several real-time PCR assays were identified for specific detection of ‘*Ca.* P. mali’ [22,23,28,29,30,31,32,33,34]. All assays were TaqMan-based and selected for evaluation. During an initial small-scale specificity test containing three ‘*Ca.* P. mali’ strains (C71, P4 and P6), two ‘*Ca.* P. prunorum’ strains (ESFY1 and P1), and one ‘*Ca.* P. pyri’ strain (P3), 16S-based assays [28,29,30,31,32,33] produced positive amplification from ‘*Ca.* P. prunorum’ strains ESFY1 and P1, while the ITS-based assay [22,23] produced positive amplification from ‘*Ca.* P. prunorum’ strain ESFY1. The C_t_ values produced by the cross-reactions were congruent with those produced by the 23S assay [19] for corresponding samples, as well as those produced by true ‘*Ca.* P. mali,’ suggesting that the observed phenomenon was a true cross-reaction and not contamination. The 16S assay relied on only two and three SNPs (single-nucleotide polymorphisms) in the probe sequence to discriminate ‘*Ca.* P. mali’ from ‘*Ca.* P. prunorum’ and ‘*Ca.* P. pyri,’ respectively. The ITS assay relied on only three SNPs in the probe sequence to discriminate ‘*Ca.* P. mali’ from ‘*Ca.* P. prunorum.’ Protocols were followed as published, so it is unknown why these cross-reactions were observed. As a result, these assays were not investigated further, but the *imp* assay was validated fully [35].

Several real-time PCR assays were identified for specific detection of ‘*Ca.* P. prunorum’ [20,21,22,23,24,26,27,56,57]. The majority of these assays were SYBR-based, and differentiation among 16SrX phytoplasmas relied on a few SNPs. Attempts were made by PPCDL to design TaqMan probes to these target regions but were unsuccessful. Three assays were TaqMan-based, with two targeting the 16S rRNA gene [24,27] and one targeting ITS [22,23] regions. The ITS assay was the same as the ‘*Ca.* P. mali’ ITS assay, except that the probe was targeting the ‘*Ca.* P. prunorum’ sequence instead of the ‘*Ca.* P. mali’ sequence; discrimination was dependent on the same three SNPs in the probe. Based on cross-reactions observed with ‘*Ca.* P. prunorum’ when using the probe specific to ‘*Ca.* P. mali,’ it was assumed that similar cross-reactions would occur with ‘*Ca.* P. mali’ when using a probe specific to ‘*Ca.* P. prunorum.’ One 16S assay took the same approach, using the assay from Baric and Dalla-Via [32] and designing a new probe to ‘*Ca.* P. prunorum’ based on the same two SNPs. As cross-reactions were observed with the 16S assay for ‘*Ca.* P. mali,’ it was assumed that discrimination by same two SNPs would also produce cross-reactions. The previously published 16S assay [27] was not considered due to the lack of information regarding specificity data for inclusivity and exclusivity tests. Additionally, this described assay is a reverse-transcriptase real-time PCR assay targeting RNA instead of DNA, making it incompatible with the DNA extraction protocol employed currently in the USA for suspected phytoplasma diagnostic samples. Therefore, no published assays were pursued for full validation.

To fulfill the agency need for a validated diagnostic for specific detection of ‘*Ca.* P. prunorum,’ draft genomes for two ‘*Ca.* P. prunorum’ strains [36] were mined for diagnostic markers. Whole-genome comparisons between ‘*Ca.* P. prunorum’ and other ‘*Ca.* Phytoplasma’ species led to the identification of the PE639 gene as a unique target. A BLAST analysis of the PE639 amplicon revealed matches to only ‘*Ca.* P. prunorum’, and validation testing demonstrated 100% specificity. Late positive amplification with PE639 was observed with two strains of ‘*Ca.* P. mali’ (AP-15 and APxN) (Appendix A). Amplicon sequencing was used to determine if the false results were from contamination. The 16S amplicons were sequenced using a MinION Flongle flow cell R10.4.1 (Oxford Nanopore Technologies, Oxford, UK) and live base calling. Quality-filtered reads were clustered against a custom database of phytoplasma genomes. Consensus sequences were built from each cluster and then classified using a BLASTn search against the same phytoplasma database. Analysis revealed the presence of ‘*Ca.* P. prunorum’ within these samples, suggesting contamination or coinfection. Contigs of ~1450 bp were assembled from these samples that matched Plum leptonecrosis phytoplasma strain LNp (GenBank accession number JQ868450) with 100% identity and 100% query coverage; query coverage was based on 1241 bp accession target length. These strains were used during initial screening of potential specific targets and primer sets. Two additional potentially specific targets were identified in this study, an 864 nt putative effector protein (designated as PE864) and a SAP09-like putative secreted protein (designated as SAP09); however, these were eliminated from further testing due to false cross-reactions with AP-15 and APxN, and only PE639 was pursued after the false contamination was identified. These two genes could be explored in the future as additional diagnostic markers.

The *imp* assay for specific detection of ‘*Ca.* P. mali’ was modified from the published version [34] and then validated fully [35]. In this study, the *imp* assay was validated for use with the CFX OPUS, which produced congruent results to the QuantStudio™ 5 and CFX Touch. Specificity was also checked against additional phytoplasmas, other pathogens commonly found on the same hosts, and healthy hosts. In Italy, ‘*Ca.* P. mali’ is present in the most important apple-growing areas, including Trentino-Alto Adige and Friuli Venezia Giulia. In the early 2000s, comparative diagnostic studies using survey samples from these regions identified strains of ‘*Ca.* P. mali’ that did not react with a monoclonal antibody-based ELISA (enzyme-linked immunosorbent assay) [58,59]. Immunoassays tend to target membrane proteins of bacteria and considering that the *imp* real-time PCR was targeting specific sequences in an outer membrane protein gene, this warranted investigation. Three of these atypical ‘*Ca.* P. mali’ strains (AP-1, AP-2, and AP-3) testing ELISA-negative were purchased from the EPPO-Q-bank at the University of Bologna for inclusion in this study. The *imp* assay demonstrated 100% specificity for ‘*Ca.* P. mali’ strains, including these three “atypical” strains, which produced positive amplification. However, while other strains produce similar *imp* and 23S C_t_ values, these atypical strains produced *imp* C_t_ values ~4–5 higher than 23S (Appendix A). Amplicon sequencing was performed as described previously to determine if the presence of other phytoplasma was responsible for the lower relative 23S C_t_ values. These samples were found to contain only ‘*Ca.* P. mali,’ confirming results and demonstrating reduced sensitivity of the *imp* assay with these atypical strains, likely due to variation in the target sequence.

A synthetic gBlocks^TM^ positive control was developed for use with all phytoplasma assays used in this study. Validation testing determined 4 fg/µL in TE buffer with 0.5 mg/mL poly(A) to be the optimal concentration, as reasonable C_t_ values were produced using this relatively low concentration, which helps prevent laboratory contamination; FAM C_t_ values of approximately 26, 32, and 27 for PE639, *imp*, and 23S, respectively, and ABY C_t_ values of 28 for all assays were observed. Stability testing over multiple rounds of freeze–thaw by multiple operators revealed extremely low variance. PCR using 16S and *tuf* primers were performed following the final freeze–thaw, and bands of appropriate size and intensity were produced. 16S amplicons produced sufficient sequence, and translation of the sequence in the appropriate reading frame revealed the unique identifier (APHIS*SCIENCE).

The design of the synthetic control facilitates its discrimination from native DNA by sequence analysis. Due to the inclusion of multiple target sequences into the control base sequence, BLAST analysis with the core nucleotide (core_nt) database returns highest matches to strawberry lethal yellows phytoplasma (CPA) str. NZSb11, complete genome (GenBank accession number CP002548), which was used for the base 16S sequence, and ‘*Candidatus* Phytoplasma australiense’ complete genome (GenBank accession number AM422018); both matched with 85.71% identity and 97% query coverage. BLAST analysis against the RefSeq Genome Database (refseq_genomes) and RefSeq Reference genomes (refseq_reference_genomes) revealed the highest matches to strain NZSb11 (GenBank accession number CP002548) with 85.71% identity and 97% query coverage and 90.5% identity and 82% query coverage, respectively. BLAST analysis against the whole-genome shotgun contigs (wgs) database returns the highest matches to various phytoplasmas with only 90% query coverage and <85% identity. When looking at graphic summary results in BLAST, alignments show gaps where target sequences were inserted and matched to the 23S sequence (usually pink in color) (Appendix A), making identifying contamination by the internal control possible by simple BLAST analysis. BLAST analysis of *tuf* nested amplicons reveal matches to the 16S sequence, which was used as the base sequence for the control; this allows quick identification of the synthetic control by BLAST analysis of *tuf* sequence. This synthetic positive control benefits screening diagnostics by reducing the need for infected reference material by testing labs, providing confidence in results through use of appropriate controls, and by allowing a single positive control to be used with all validated PCR-based assays for phytoplasmas included in this study.

## 5. Conclusions

Overall, the PE639 and *imp* assays provide 100% specificity for ‘*Ca.* P. prunorum’ and ‘*Ca.* P. mali,’ respectively, along with high repeatability, reproducibility, and precision, comparable linearity and sensitivity to each other and the 23S assay, and reasonable price points. The cost per reaction for the PE639 and *imp* assays was calculated to be ~USD 0.98 and ~USD 0.92, respectively, based on the reagents and oligonucleotides used per reaction, plus a 20% increase to account for disposable supplies like tips, tubes, and gloves; this does not reflect operational costs. The cost of using the synthetic gBlocks^TM^ positive control was also calculated. Based on rehydration and dilution to 4 fg/µL in 0.5 mg/mL poly(A) in TE buffer, the cost for every 1000 reactions was calculated to be ~USD 3.52; the largest contributing factor to cost was poly(A) (Appendix A). The modified *imp* assay improves upon the published version [34] by including a plant internal control primer set, allowing quality control assessment of individual reactions [35]. The PE639, *imp*, and 23S assays can be run in parallel, as all assays were optimized for reactions to run under the exact same parameters, allowing for general detection of phytoplasmas and discrimination for ‘*Ca.* P. mali’ and ‘*Ca.* P. prunorum’ to happen during the same run, if needed. Additionally, the LoDs of all assays were congruent, and real-time results were confirmed by cPCR and sequencing. As with all assays, performance should be continuously monitored, especially regarding specificity, as new strains are known to arise over time.

## Figures and Tables

**Figure 1 microorganisms-13-00929-f001:**
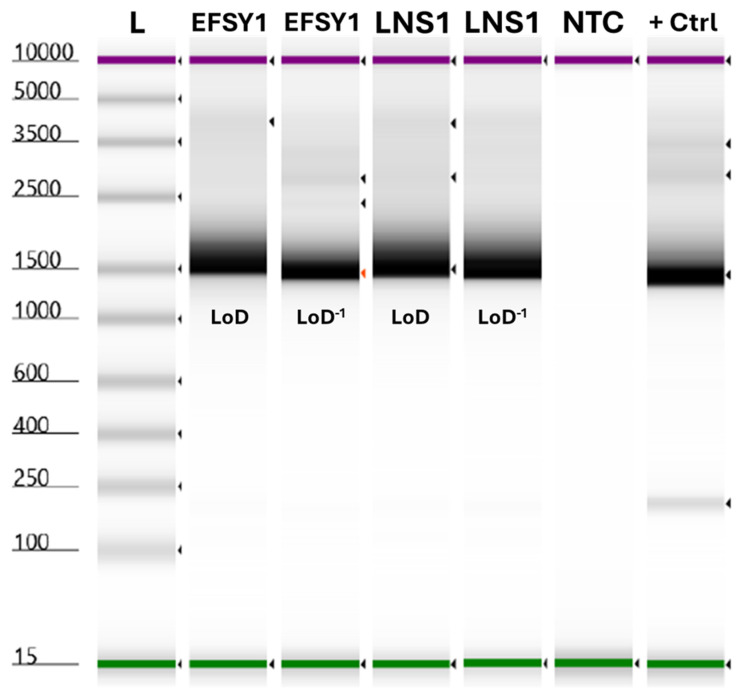
Electronic gel image showing amplicons resulting from a semi-nested PCR using 16S primers for phytoplasmas. Two dilutions (limit of detection (LoD) and LoD^−1^) of DNA from ‘*Candidatus* Phytoplasma prunorum’ strains ESFY1 and LNS1 in healthy peach (*Prunus persica*) DNA and synthetic gBlocks™ positive control (4 fg/µL) were subjected to semi-nested 16S PCR amplification; + Ctrl: 4 fg/µL synthetic control; NTC: no-template control. Amplicons were analyzed using the 4200 TapeStation System, D5000 ScreenTape, D5000 DNA Ladder (L), and D5000 Sample Buffer according to the manufacturer’s instructions. The data show bands of appropriate size (~1500 bp) were produced for all samples tested, as well as the synthetic control; no bands were produced for the NTC.

**Figure 2 microorganisms-13-00929-f002:**
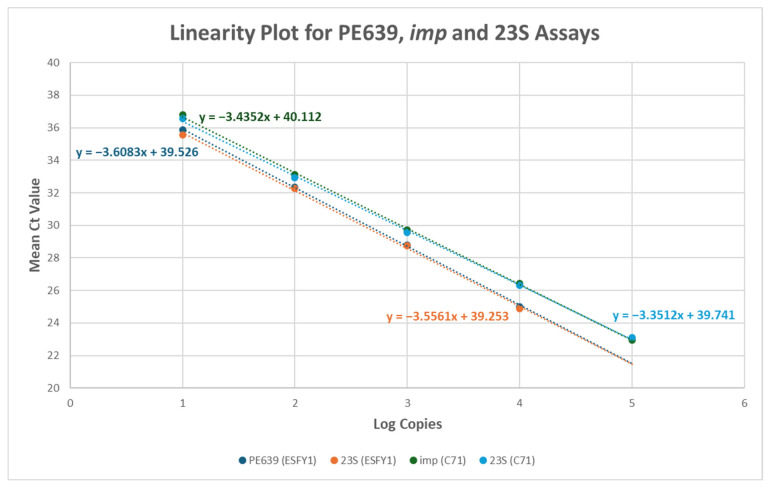
Linear amplification plots for 10-fold serial dilutions of DNA from ‘*Candidatus* Phytoplasma prunorum’ strain ESFY1 and ‘*Ca.* P. mali’ strain C71. Strains EFSY1 and C71 were each tested with two assays: PE639 and 23S for ESFY1, and *imp* and 23S for C71. Data points represent mean C_t_ values from three independent operators and two instruments (QuantStudio™ 5 Real-Time PCR System and CFX96 OPUS Real-Time PCR System) (N = 12 per data point). Linear regression equations are displayed.

**Table 1 microorganisms-13-00929-t001:** Target and non-target organisms tested in this study.

Organism	RFLP Group	Sample ID	Host ^a^	Host (Common)	Origin	Source ^b,c^
*Alternaria alternata*	N/A	22033001-01 (B)	*Citrus limon*	Lemon	USA, TX	PPCDL
*Aureobasidium pullulans*	N/A	DSM 14940	*Malus domestica*	Apple	Germany	Westbridge Ag
*Aureobasidium pullulans*	N/A	DSM 14941	*Malus domestica*	Apple	Germany	Westbridge Ag
‘*Candidatus* Phytoplasma’	16SrII-C	CoPh	*Cocos nucifera*	Coconut	Tanzania	EPPO-Q-bank
‘*Candidatus* Phytoplasma’	16SrIII-B	GY-U	*Vitis vinifera* var. ‘Chardonnay’	Chardonnay grape	Italy	EPPO-Q-bank
‘*Candidatus* Phytoplasma’	16SrIX-C	PEY	*Pichris echioides*	Bristly oxtongue	Italy	EPPO-Q-bank
‘*Candidatus* Phytoplasma’	16SrV-C	ALY	*Alnus* sp.	Alder	Italy	EPPO-Q-bank
‘*Candidatus* Phytoplasma’	16SrXI-C	BVK	*Psammotettix cephalotes*	Leafhopper	Germany	EPPO-Q-bank
‘*Candidatus* Phytoplasma asteris’	16SrI-A	23092601-01	*Vitis* spp.	Grape	MN, USA	PPCDL
‘*Candidatus* Phytoplasma asteris’	16SrI-A	23092601-02	*Vitis* spp.	Grape	MN, USA	PPCDL
‘*Candidatus* Phytoplasma asteris’	16SrI-A	23092601-03	*Vitis* spp.	Grape	MN, USA	PPCDL
‘*Candidatus* Phytoplasma asteris’	16SrI-F	AY-A	*Prunus armeniaca*	Apricot	Spain	EPPO-Q-bank
‘*Candidatus* Phytoplasma aurantifolia’	16SrII-B	WBDL	*Citrus* spp.	Lime	Oman	EPPO-Q-bank
‘*Candidatus* Phytoplasma brasilliense’	16SrXV-A	SuV	*Hibiscus* spp.	Hibiscus	France	EPPO-Q-bank
‘*Candidatus* Phytoplasma mali’	16SrX-A	AP-1	*Malus domestica*	Apple	Italy	EPPO-Q-bank
‘*Candidatus* Phytoplasma mali’	16SrX-A	AP-15	*Malus domestica*	Apple	Italy	EPPO-Q-bank
‘*Candidatus* Phytoplasma mali’	16SrX-A	AP-2	*Malus domestica*	Apple	Italy	EPPO-Q-bank
‘*Candidatus* Phytoplasma mali’	16SrX-A	AP-3	*Malus domestica*	Apple	Italy	EPPO-Q-bank
‘*Candidatus* Phytoplasma mali’	16SrX-A	APxN	*Malus domestica*	Apple	Italy	EPPO-Q-bank
‘*Candidatus* Phytoplasma mali’	16SrX-A	C71	*Malus domestica*	Apple	Italy	K. Zikeli
‘*Candidatus* Phytoplasma mali’	16SrX-A	P4	*Malus domestica*	Apple	Slovenia	N. Mehle
‘*Candidatus* Phytoplasma mali’	16SrX-A	P6	Pooled *Malus domestica/Cacopsylla picta*	Apple/psyllid (jumping plant lice)	Slovenia	N. Mehle
‘*Candidatus* Phytoplasma palmae’	16SrIV-D	19101702-01	*Phoenix sylvestris*	Wild date palm	MS, USA	PPCDL
‘*Candidatus* Phytoplasma palmae’	16SrIV-D	19110501-02	*Phoenix sylvestris*	Wild date palm	MS, USA	PPCDL
‘*Candidatus* Phytoplasma pini’-related	16SrXXI-B	MDPP	*Pinus pungens*	Mountain pine	MD, USA	S. Costanzo
‘*Candidatus* Phytoplasma pruni’ rrnA	16SrIII-A rrn	19122001-22	*Prunus* spp.	-	WA, USA	PPCDL
‘*Candidatus* Phytoplasma pruni’ rrnA	16SrIII-A rrn	19122001-41	*Prunus* spp.	-	WA, USA	PPCDL
‘*Candidatus* Phytoplasma pruni’ rrnA	16SrIII-A rrn	20012301-04	*Prunus* spp.	Cherry	WA, USA	PPCDL
‘*Candidatus* Phytoplasma pruni’ rrnA	16SrIII-A rrn	20012301-05	*Prunus* spp.	Cherry	WA, USA	PPCDL
‘*Candidatus* Phytoplasma pruni’ rrnA	16SrIII-A rrn	20012301-06	*Prunus* spp.	Cherry	WA, USA	PPCDL
‘*Candidatus* Phytoplasma pruni’ rrnA	16SrIII-A rrn	20012301-07	*Prunus* spp.	Cherry	WA, USA	PPCDL
‘*Candidatus* Phytoplasma pruni’ rrnA	16SrIII-A rrn	20012301-08	*Prunus* spp.	Cherry	WA, USA	PPCDL
‘*Candidatus* Phytoplasma pruni’ rrnA	16SrIII-A rrn	20012301-09	*Prunus* spp.	Cherry	WA, USA	PPCDL
‘*Candidatus* Phytoplasma pruni’ rrnA	16SrIII-A rrn	20012301-10	*Prunus* spp.	Cherry	WA, USA	PPCDL
‘*Candidatus* Phytoplasma pruni’ rrnA	16SrIII-A rrn	20012301-11	*Prunus* spp.	Cherry	WA, USA	PPCDL
‘*Candidatus* Phytoplasma pruni’ rrnA	16SrIII-A rrn	21021201-02	*Prunus* spp.	-	WA, USA	PPCDL
‘*Candidatus* Phytoplasma pruni’ rrnA	16SrIII-A rrn	21021201-03	*Prunus* spp.	-	WA, USA	PPCDL
‘*Candidatus* Phytoplasma prunorum’	16SrX-F	ESFY1	*Prunus salicina*	Japanese/Chinese plum	Italy	K. Zikeli
‘*Candidatus* Phytoplasma prunorum’	16SrX-F	ESFY-1A	*Prunus armeniaca*	Apricot	Italy	EPPO-Q-bank
‘*Candidatus* Phytoplasma prunorum’	16SrX-F	ESFY-1P	*Prunus salicina*	Japanese plum	Italy	EPPO-Q-bank
‘*Candidatus* Phytoplasma prunorum’	16SrX-F	ESFY-1PE	*Prunus persica*	Peach	Italy	EPPO-Q-bank
‘*Candidatus* Phytoplasma prunorum’	16SrX-F	ESFY-2A	*Prunus armeniaca*	Apricot	Italy	EPPO-Q-bank
‘*Candidatus* Phytoplasma prunorum’	16SrX-F	ESFY-2P	*Prunus salicina*	Japanese plum	Italy	EPPO-Q-bank
‘*Candidatus* Phytoplasma prunorum’	16SrX-F	ESFY-2PE	*Prunus persica*	Peach	Italy	EPPO-Q-bank
‘*Candidatus* Phytoplasma prunorum’	16SrX-F	LNp	*Prunus salicina*	Japanese plum	Italy	EPPO-Q-bank
‘*Candidatus* Phytoplasma prunorum’	16SrX-F	LNS1	*Prunus salicina*	Japanese plum	Italy	EPPO-Q-bank
‘*Candidatus* Phytoplasma prunorum’	16SrX-F	LNS2	*Prunus salicina*	Japanese plum	Italy	EPPO-Q-bank
‘*Candidatus* Phytoplasma prunorum’	16SrX-F	P1	*Prunus americana* and *P. persica*	American plum and peach	Slovenia	N. Mehle
‘*Candidatus* Phytoplasma pyri’	16SrX-C	20012301-03	*Pyrus* spp.	Pear	WA, USA	PPCDL
‘*Candidatus* Phytoplasma pyri’	16SrX-C	21021201-01	*Pyrus* spp.	Pear	WA, USA	PPCDL
‘*Candidatus* Phytoplasma pyri’	16SrX-C	23111702-01	*Pyrus* spp.	Pear	WA, USA	PPCDL
‘*Candidatus* Phytoplasma pyri’	16SrX-C	P3	*Pyrus communis*	European/common pear	Slovenia	N. Mehle
‘*Candidatus* Phytoplasma pyri’	16SrX-C	PD	*Pyrus communis*	European/common pear	Germany	EPPO-Q-bank
‘*Candidatus* Phytoplasma rubi’	16SrV-E	RuS	*Rubus* spp.	Brambles	Italy	EPPO-Q-bank
‘*Candidatus* Phytoplasma solani’	16SrXII-A	CH-1	*Vitis vinifera* var. ’Chardonnay’	Chardonnay grape	Italy	EPPO-Q-bank
‘*Candidatus* Phytoplasma solani’	16SrXII-A	LNIV	*Prunus salicina*	Japanese plum	Italy	EPPO-Q-bank
‘*Candidatus* Phytoplasma vitis’-related	16SrV-C	19041105-01	*Alnus* spp.	Alder	WA, USA	PPCDL
‘*Candidatus* Phytoplasma vitis’-related	16SrV-C	19081401-01	*Alnus* spp.	Alder	WA, USA	PPCDL
‘*Candidatus* Phytoplasma vitis’-related	16SrV-C	19122001-01	*Alnus* spp.	Alder	WA, USA	PPCDL
‘*Candidatus* Phytoplasma vitis’-related	16SrV-C	19122001-26	*Alnus* spp.	Alder	WA, USA	PPCDL
‘*Candidatus* Phytoplasma vitis’-related	16SrV-C	19122001-32	*Alnus* spp.	Alder	WA, USA	PPCDL
‘*Candidatus* Phytoplasma vitis’-related	16SrV-C	19122001-40	*Prunus* spp.	-	WA, USA	PPCDL
‘*Candidatus* Phytoplasma vitis’-related	16SrV-C	19122001-42	*Alnus* spp.	Alder	WA, USA	PPCDL
‘*Candidatus* Phytoplasma vitis’-related	16SrV-C	21012601-01	*Alnus* spp.	Alder	WA, USA	PPCDL
‘*Candidatus* Phytoplasma vitis’-related	16SrV-C	21012601-02	*Alnus* spp.	Alder	WA, USA	PPCDL
‘*Candidatus* Phytoplasma vitis’-related	16SrV-C	21031201-01	*Malus* spp.	Apple	WA, USA	PPCDL
‘*Candidatus* Phytoplasma vitis’-related	16SrV-C	21031201-02	*Malus* spp.	Apple	WA, USA	PPCDL
‘*Candidatus* Phytoplasma vitis’-related	16SrV-C	21042602-01	*Malus* spp.	Apple	WA, USA	PPCDL
‘*Candidatus* Phytoplasma vitis’-related	16SrV-C	21042602-02	*Malus* spp.	Apple	WA, USA	PPCDL
*Colletotrichum gloeospiriodes*	N/A	22033001-01 (C)	*Citrus limon*	Lemon	USA, TX	PPCDL
*Colletotrichum gloeospiriodes*	N/A	22033001-01 (D1)	*Citrus limon*	Lemon	USA, TX	PPCDL
*Colletotrichum queenslandicum*	N/A	17030501 (C-11)	*Citrus limon*	Lemon	USA, TX	PPCDL
*Colletotrichum queenslandicum*	N/A	17030501 (C-5)	*Citrus limon*	Lemon	USA, TX	PPCDL
*Erwinia amylovora*	N/A	Ea110	*Malus domestica*	Apple	USA, OR	V. Stockwell
*Erwinia amylovora*	N/A	Ea153	*Malus domestica*	Apple	USA, OR	V. Stockwell
*Erwinia amylovora*	N/A	Fire blight	*Malus domestica*	Apple	USA, MD	J. Yasuhara-Bell
*Erwinia amylovora*	N/A	Fire blight	*Malus domestica*	Apple	USA, MD	PGQP
*Erwinia amylovora*	N/A	Parkdale	*Malus domestica*	Apple	USA, MI	V. Stockwell
*Erwinia aphidicola*	N/A	17110702-04	*Cucumis melo* var. ’mlada’	Mlada melon	USA, CA	PPCDL
*Erwinia billingae*	N/A	Eh24	*Pyrus* spp.	Pear	Turkey	V. Stockwell
*Erwinia billingae*	N/A	NCPPB 661T	*Pyrus communis*	European/common pear	UK	NCPPB
*Erwinia persicina*	N/A	19122706-01	*Cucumis melo*	Melon	USA, CA	PPCDL
*Erwinia persicina*	N/A	LA611	*Rubus* spp.	Raspberry	USA, WA	V. Stockwell
*Erwinia persicina*	N/A	LA659	*Rubus* spp.	Raspberry	USA, WA	V. Stockwell
*Erwinia piriflorinigrans*	N/A	CFBP 5888T	*Pyrus communis*	European/common pear	Spain	CFPB
*Erwinia pyrifoliae*	N/A	23-02275	*Fragaria* × *ananassa*	Strawberry	USA, OH	PPCDL
*Erwinia pyrifoliae*	N/A	Ejp 556	*Pyrus pyrifolia*	Asian Pear	Japan	A. Svircev
*Erwinia pyrifoliae*	N/A	Ejp 617	*Pyrus pyrifolia*	Asian Pear	Japan	A. Svircev
*Erwinia pyrifoliae*	N/A	Ep 28/96	*Pyrus pyrifolia*	Asian Pear	Korea	A. Svircev
*Erwinia pyrifoliae*	N/A	Ep 4/97	*Pyrus pyrifolia*	Asian Pear	Korea	A. Svircev
*Erwinia rhapontici*	N/A	19122702-01	*Delphinium* spp.	Larkspur	USA, CA	PPCDL
*Erwinia tasmaniensis*	N/A	Et1/99T	*Malus domestica*	Apple	Australia	V. Stockwell
*Erwinia uzenensis*	N/A	NCPPB 4475T	*Pyrus communis*	European/common pear	Japan	NCPPB
*Monilinia fructicola*	N/A	19090401-01A	*Prunus persica*	Peach	USA, WV	K. Zeller
*Monilinia fructicola*	N/A	19090401-02A	*Malus domestica*	Apple	USA, MN	PPCDL
*Monilinia fructicola*	N/A	Mf35	*Malus domestica*	Apple	USA, MN	PPCDL
*Monilinia fructigena*	N/A	Mfg2-GE-A E	*Malus domestica*	Apple	Hungary	K. Zeller
*Monilinia laxa*	N/A	PSG1	*Prunus persica*	Peach	Italy	K. Zeller
*Monilinia polystroma*	N/A	SP61	*Prunus* spp.	Plum	Poland	K. Zeller
*Neofabraea alba*	N/A	PD-1696	*Malus domestica*	Apple	USA, WA	T. Wilson
*Neofabraea* sp.	N/A	PD-1597	*Malus domestica*	Apple	USA, WA	T. Wilson
*Neofabraea* sp.	N/A	PD-1617	*Malus domestica*	Apple	USA, WA	T. Wilson
*Neofabraea* sp.	N/A	PD-1655B	*Malus domestica*	Apple	USA, WA	T. Wilson
*Pantoea agglomerans*	N/A	23091801-01	*Triticum aestivum*	Wheat	USA, CO	PPCDL
*Pantoea agglomerans*	N/A	23091801-03	*Triticum aestivum*	Wheat	USA, CO	PPCDL
*Pantoea agglomerans*	N/A	23091801-04	*Triticum aestivum*	Wheat	USA, CO	PPCDL
*Pantoea agglomerans*	N/A	E325	*Malus domestica*	Apple	USA, WA	V. Stockwell
*Pantoea allii*	N/A	23091801-02	*Triticum aestivum*	Wheat	USA, CO	PPCDL
*Pantoea ananatis*	N/A	CES-14 (SM-272)	-	-	Philippines	S. Miller
*Pantoea ananatis*	N/A	CES-5 (JM-55)	-	Soil	Philippines	S. Miller
*Pantoea ananatis*	N/A	E	*Oryza sativa*	Rice	USA, AR	J. Leach; E. Peachey
*Pantoea stewartii*	N/A	DCop3-07	*Zea mays*	Corn	-	S. Miller; D. Coplin
*Pantoea stewartii*	N/A	PP685	-	-		S. Miller
*Pantoea vagans*	N/A	C9-1	*Malus domestica*	Apple	USA, MI	V. Stockwell
*Phacidiopycnis wasingtonensis*	N/A	PD-1597	*Malus domestica*	Apple	USA, WA	T. Wilson
*Phacidiopycnis wasingtonensis*	N/A	PD-1655B	*Malus domestica*	Apple	USA, WA	T. Wilson
*Pseudomonas fluorescens*	N/A	A506	*Pyrus* spp.	Pear	USA, CA	V. Stockwell
*Pseudomonas syringae*	N/A	JL2583	*Vaccinium* spp.	Blueberry	USA, OR	V. Stockwell
*Sphaeropsis pyriputrescens*	N/A	PD-1655B	*Malus domestica*	Apple	USA, WA	T. Wilson
*Venturia inaequalis*	N/A	Apple Scab	*Malus domestica*	Apple	USA, MD	PGQP
*Xanthomonas arboricola* pv. *corylina*	N/A	JL2611	*Corylus avellana*	Hazelnut	USA, OR	V. Stockwell
*Xyllela fastidiosa*	N/A	XFS 253	*Salix* spp.	Willow	USA, WA	T. Wilson
*Xyllela fastidiosa*	N/A	XFS 254	*Lonicera* spp.	Honeysuckle	USA, WA	T. Wilson
*Xyllela fastidiosa*	N/A	XFS 946	*Rosa* spp.	Rose	USA, WA	T. Wilson
*Xyllela fastidiosa* subsp. *multiplex*	N/A	Peach Leaves Georgia	*Prunus persica*	Peach	USA, GA	Y.K. Jo
*Xyllela fastidiosa* subsp. *multiplex*	N/A	Peach Texas A&M (1)	*Prunus persica*	Peach	USA, TX	Y.K. Jo

^a^ Host designations for phytoplasmas represent the original infected host plant. All phytoplasmas were propagated in *Catharanthus roseus* (Madagascar periwinkle) prior to DNA extraction, except for diagnostic samples, three ‘*Ca*. P. mali’ (AP-1, AP-2, and AP-3) and six ‘*Ca*. P. prunorum’ (ESFY-1A, ESFY-2A, ESFY-1P, ESFY-2P, ESFY-1PE, and ESFY-2PE) strains, which were from their original hosts. ^b^ A. Svircev—Agriculture and Agri-Food Canada; CFBP—French Collection for Plant-Associated Bacteria; D. Coplin—The Ohio State University Department of Plant Pathology; E. Peachey (Luna)—Colorado State University Department of Agricultural Biology; EPPO-Q-bank—European and Mediterranean Plant Protection Organization Q-bank; J. Leach—Colorado State University Department of Agricultural Biology; J. Yasuhara-Bell—USDA APHIS PPQ S&T PPCDL; K. Zeller—USDA APHIS PPQ S&T PPCDL; K. Zikeli—Julius Kühn-Institut Federal Research Institute for Cultivated Plants, Institute for Plant Protection in Fruit Crops and Viticulture, Dossenheim, Germany; N. Mehle—National Institute of Biology, Department of Biotechnology and Systems Biology, Ljubljana, Slovenia; NCPPB—National Collection of Plant Pathogenic Bacteria, Fera, UK; PGQP—USDA APHIS Plant Germplasm Quarantine Program; PPCDL—original sample was received for diagnostic testing by the USDA APHIS PPQ S&T PPCDL; S. Costanzo—USDA APHIS PPQ S&T PPCDL; S. Miller—The Ohio State University Department of Plant Pathology; T. Wilson—Washington State Department of Agriculture; V. Stockwell—USDA ARS Corvallis, OR; Westbridge Ag—Westbridge Agricultural Products; Y.K. Jo—Texas A&M University, Department of Plant Pathology and Microbiology. ^c^ Phytoplasma strains from PPCDL represent domestic samples collected within the United States that tested positive for ‘*Candidatus* Phytoplasma’ by the 23S rRNA assay [19]. Species and 16Sr group identification resulted from 16S rRNA sequencing and analysis using *i*PhyClassifier [37].

**Table 2 microorganisms-13-00929-t002:** Oligonucleotide primers and probes used in this study.

Target Organism	Primer	Primer Sequence (5′->3′)	Final Concentration (nM)	Gene Target	Amplicon (bp)	Ref.
‘*Candidatus* Phytoplasma prunorum’	ESFY-PE639-F	ACAGGCCGCGAATTTATTACT	200	Putative effector (639 nt)	139	This study
	ESFY-PE639-R	GGACCGATGCTTTCACTGTT	200			
	ESFY-PE639-P	FAM-AGATCAACGCGTACAAGCAACAGA-QSY	100			
‘*Candidatus* Phytoplasma mali’	fimpAP	GGTTCAGTTGTTGGTGCTT	100	*imp*	88	[34,35]
	rimpAP	TTTSTTGTTTACTTTKTGATGAAA	500			
	qimpAP	FAM-AGGCCAGAAACTAATAGACCAAGCT-QSY	100			
‘*Candidatus* Phytoplasma’	JH-F 1	GGTCTCCGAATGGGAAAACC	240	23S	142–149	[19]
	JH-F all	ATTTCCGAATGGGGCAACC	240			
	JH-R	CTCGTCACTACTACCRGAATCGTTATTAC	240			
	JH-P uni	FAM-AACTGAAATATCTAAGTAAC-MGB-NFQ	120			
Plant host	F	GACTACGTCCCTGCCCTTTG	48	18S	68	[13]
	R-mod ^a^	AACAYTTCACCGGAYCATTCA	48			
	Probe ^b^	ABY-ACACACCGCCCGTCGCTCC-QSY	24			
‘*Candidatus* Phytoplasma’	P1	AAGAGTTTGATCCTGGCTCAGGATT	400	16S	1538 ^c^	[53,54]
	P1A	AACGCTGGCGGCGCGCCTAATAC	400			
	16S-SR	GGTCTGTCAAAACTGAAGATG	400			

^a^ Primer modified from original sequence to include degenerate bases. ^b^ For use with a Bio-Rad CFX96 OPUS, the 18S (ABY-QSY) fluorophore/quencher pair was changed to VIC-QSY and Cy3-IAbRQSp when duplexed with PE639 (‘*Ca.* P. prunorum’) and *imp* (‘*Ca.* P. mali’), respectively. ^c^ Amplicon size following 2nd round of semi-nested PCR.

## Data Availability

All relevant data are provided in the main document and Appendix A.

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
