# Peer review of "Genome-Informed Real-Time PCR Assay for Detection of ‘Candidatus Phytoplasma Prunorum,’ Which Is Associated with European Stone Fruit Yellows"

_microorganisms, 2025, doi:10.3390/microorganisms13040929_

Round 1

Reviewer 1 Report

Comments and Suggestions for Authors

The authors present a new real-time PCR assay for the detection of Ca. P. prunorum. The assay is evaluated using gBlocks and a huge range of material. The aim is to fulfil the requirements for a diagnostic assay in terms of sensitivity, specificity and reliability. Unfortunately, the manuscript has some weaknesses that need to be addressed before a final decision. These include the consideration of an existing assay, the unfavourable compilation of the table contents, the lack of necessary information on sensitivity (copy number) and the gaps in the material and methods section, which could be better separated from/synchronised with the results (see subheadings).
A summary of the criticisms follows below, with aspects that are of particular importance for the study marked as ‘major’.

(MAJOR) Introduction and discussion, 
did not take into account the publication by Minguzzi et al. 2016 on the detection of Ca. P. prunorum in qPCR (https://doi.org/10.1371/journal.pone.0146515).

Critics on the introduction,

please provide more information on ESFY, this is necessary to clarify the need for further assays.

(MAJOR) Material and methods, 
Table 1 is not reasonable and meaningful in the current form:

an overview should be given in the M+M section and the information should be presented with the results in the Results section;

abbreviations should be introduced/used consistently;

the formatting should be compact;

strains must always be named;

a distinction between pure cultures and plant samples should be made clear;

the category ‘host (common)’ should be skipped (if necessary, mention it in the text);

page breaks have to be adequately formatted;

the respective DNA extraction method should be indexed;

the template concentrations, as well as the qPCR experiments and the CT values obtained should be added (merged with S6/7)

=> If necessary, consider shortening the table in the manuscript. 

Critics, material and methods (L94-98),
A specific description of the methods and their modifications must be provided here.

(MAJOR) Material and methods (L149-54),
Identification of PE639 is a result (wrong section, result), information on finding the gene is missing (gene ID, locus tag etc.); reference for the assignment as (putative) effector is missing.

(MAJOR) Material and methods,
BLAST-parameter are critical for this approach and have to be noted (e.g. word size, filter etc.)

Critics, material and methods (L169),
Please explain what ‘modified’ means here, especially since it is addressed again in the results.

Critics, material and methods (L173-5),
Please explain in detail or discard.

Critics, material and methods (L177),
Explain duplex

(MAJOR) Material and methods, section 2.6, 
Sensitivity testing on gBlocks is missing (copy number). 

(MAJOR) Results, section 3.1,
It is not clear to what extent the sensitivity of the assay is analysed here, since neither known concentrations for the template or pathogens or gBlocks are provided, nor is relative quantification carried out.

Critics, results, section 3.2,
it remains unclear in the text what the results refer to cPCR or qPCR. Table 1 and S6/7 should be merged. 

(MAJOR) Discussion, L438-54,
it remains unclear what is presented here. Results without description in the M+M section?

Critics, discussion, L479,
On what basis is a low concentration mentioned here? Compared to?

Critics, discussion, L484,
High-quality sequence? Skip or provide quality values.

Critics, conclusion, L518,
Calculation cannot be reproduced, skip or explain.

(MAJOR) running gag,
please do not use 16S alone, please change to 16S rRNA (for the gene) or 16S rDNA (template)

Critics, Figure S1
Colour choice hinders reading in print.

Critics, Figure S5
This is a screenshot and should be described as such.

Reviewer 2 Report

Comments and Suggestions for Authors

This paper reports on the development of a real-time PCR assay for specific detection of ‘Candidatus Phytoplasma Prunorum’ based on draft genome sequences of two strains (ESFY1 and LNS1) of the mentioned pathogen, which are available in the GenBank public database. Whole genome comparisons of ESFY1 and LNS1 strains led to identification of a diagnostic marker gene, the PE639 gene, which was specific for ‘Candidatus Phytoplasma prunorum’. Since ‘Candidatus Phytoplasma Prunorum’ is not yet established in American continent, the newly developed real-time PCR assay was intended as a suitable tool for surveillance and detection efforts to safeguard US agriculture and facilitate safe trade. The paper contains new information which may be of some interest to the phytoplasma reseach community. However, the following points should be corrected and/or improved before acceptance:

Throughout the paper. A substantial numbers of field-collected infected samples originating from different geographic areas and from different Prunus species needs to be examined in order to prove that the newly developed diagnostic tool is able to amplify all strains of the ‘Candidatus Phytoplasma prunorum’. In the present work, only a few field-collected samples provided by A. Bertaccini (University of Bologna, Italy) were examined. These samples most probably originated from a rescricted area of northern Italy.

Material and Method section and Table 1. In order to get clear information on the identity and taxonomic assignment on the phytoplasma strains examined in the present work, appropriate references should be provided for each of them.

Line 93. It is not clear what “Ca. Phytoplasma’-positive diagnostic samples” does it mean. Are these samples collected exclusively from Prunus species in U.S. countries? Please clarify.

Table 1. Among the tested phytoplasma strains it should be clearly shown which are periwinkle-maintained strains and which are field-collected strains. It is well known that some of the periwinkle-maintained strains are not detectable in naturally-infected stone fruit trees.

Material and Method section and Table 1. Authors should explain why they tested and compared a series of unrelated pathogens such as various fungal and bacterial agents.

Table 1. ‘Candidatus Phytoplasma vitis’ and ‘Candidatus Phytoplasma alni’ are not officially described as a new taxonomic entities. Therefore, these names should be avoided.

Material and Method section and Table 1. I suggest to move Table 1 to Supplementary Materials.

Round 2

Reviewer 1 Report

Comments and Suggestions for Authors

To the editor and authors,

A sufficient number of changes have been made in response to the criticism. Table 1 is still considered visually unsuccessful, but this is irrelevant to the content. Only one wording needs to be changed before acceptance as the statement in the current wording is not well chosen/misleading,

L484-7: „The other 16S assay by Minguzzi et al. 2016 was not considered, as information regarding the validation of the assay itself was not present in the manuscript. Lack of specificity data for inclusivity and exclusivity tests made it difficult assess the performance of the assay.”

A more neutral wording should be provided, especially considering that the Yasuhara-Bell & Rivera manuscript did not perform a comparison of assays, used templates without DNA measurement, and performed a BLAST comparison with standard parameters. A possible phrasing would be:

“The previously published 16S assay by Minguzzi et al. 2016 was not considered due to the lack of information on specificity data for the inclusivity and exclusivity tests.”

Author Response

Response to Reviewer 1 Comments

1. Summary

2. Questions for General Evaluation

Reviewer’s Evaluation

Response and Revisions

Does the introduction provide sufficient background and include all relevant references?

Can be improved

See the point-by-point response to reviewer comments

Is the research design appropriate?

Can be improved

Are the methods adequately described?

Yes

Are the results clearly presented?

Yes

Are the conclusions supported by the results?

Must be improved

3. Point-by-point response to Comments and Suggestions for Authors

Comments 1: A sufficient number of changes have been made in response to the criticism. Table 1 is still considered visually unsuccessful, but this is irrelevant to the content. Only one wording needs to be changed before acceptance as the statement in the current wording is not well chosen/misleading,

L484-7: „The other 16S assay by Minguzzi et al. 2016 was not considered, as information regarding the validation of the assay itself was not present in the manuscript. Lack of specificity data for inclusivity and exclusivity tests made it difficult assess the performance of the assay.”

A more neutral wording should be provided, especially considering that the Yasuhara-Bell & Rivera manuscript did not perform a comparison of assays, used templates without DNA measurement, and performed a BLAST comparison with standard parameters. A possible phrasing would be:

“The previously published 16S assay by Minguzzi et al. 2016 was not considered due to the lack of information on specificity data for the inclusivity and exclusivity tests.”

Response 1: Thank you for pointing this out. The text was changed to read “The previously published 16S assay by Minguzzi et al. 2016 was not considered due to the lack of information regarding specificity data for inclusivity and exclusivity tests.”

4. Response to Comments on the Quality of English Language

Point 1: No issue identified

5. Additional clarifications

No additional clarifications

Reviewer 2 Report

Comments and Suggestions for Authors

The revised version is suitable for publication in its current for.

Author Response

Response to Reviewer 2 Comments

1. Summary

2. Questions for General Evaluation

Reviewer’s Evaluation

Response and Revisions

Does the introduction provide sufficient background and include all relevant references?

Yes

See the point-by-point response to reviewer comments

Is the research design appropriate?

Yes

Are the methods adequately described?

Yes

Are the results clearly presented?

Yes

Are the conclusions supported by the results?

Yes

3. Point-by-point response to Comments and Suggestions for Authors

Comments 1: The revised version is suitable for publication in its current for.

Response 1: No response necessary.

4. Response to Comments on the Quality of English Language

Point 1: No issue identified

5. Additional clarifications

We thank the reviewer for taking the time to review the manuscript once again and for their assessment that the manuscript is suitable for publication.